# Serum Cholesterol Concentration on Admission in 415 Dogs Envenomated by *Daboia (Vipera) palaestinae* as a Marker of Envenomation Severity and Outcome—A Retrospective Study

**DOI:** 10.3390/toxins15100609

**Published:** 2023-10-12

**Authors:** Sigal Klainbart, Efrat Kelmer, Iris Beeri-Cohen, Yael Keinan, Gilad Segev, Itamar Aroch

**Affiliations:** 1Department of Small Animal Emergency and Critical Care, Koret School of Veterinary Emergency and Critical Care, The Hebrew University of Jerusalem, 229 Herzel St., P.O. Box 12, Rehovot 7610001, Israel; efrat.kelmer@mail.huji.ac.il (E.K.); cohen.iris@mail.huji.ac.il (I.B.-C.); 2Department of Small Animal Internal Medicine, Koret School of Veterinary Emergency and Critical Care, The Hebrew University of Jerusalem, 229 Herzel St., P.O. Box 12, Rehovot 7610001, Israel; yael.keinan@mail.huji.ac.il (Y.K.); gilad.segev@mail.huji.ac.il (G.S.); itamar.aroch@mail.huji.ac.il (I.A.)

**Keywords:** snakebite, venom, canine, viper, snake, platelet count, creatinine, albumin, hypocholesterolemia

## Abstract

*Daboia* (*Vipera*) *palaestinae* (*Dp*), accounts for most envenomations in humans and dogs in Israel. In humans envenomed by *Dp*, serum cholesterol concentration (sChol) is inversely correlated with envenomation severity. This study examined the utility of sChol upon admission in dogs envenomed by *Dp* as an envenomation severity and outcome marker. Data upon admission, including sChol, were retrospectively collected from the medical records of dogs with proven *Dp* envenomation. The study included 415 dogs. The mortality rate was 11%. The heart rate upon admission was higher in non-survivors than in survivors. Signs of bleeding or hematoma and circulatory shock signs were more frequent among non-survivors compared to survivors. sChol, the platelet count, and serum albumin concentration (sAlb) were lower, while serum creatinine concentration was higher among non-survivors. sChol and sAlb were moderately, positively, and significantly correlated. sChol was significantly, negatively, albeit weakly, correlated with the length of hospitalization and the heart rate. sChol was lower in dogs admitted >12 h post-envenomation than in those admitted later. In dogs, sChol upon admission is a potential marker of severity and outcome of *Dp* envenomation. The platelet count, sAlb, and sCreat might also be potential markers.

## 1. Introduction

*Daboia (Vipera) palaestinae* (*Dp*) is the most common venomous snake in Israel, accounting for several hundred envenomations in humans and domestic animals annually [1,2,3,4,5,6], with mortality rates of 0.5–2% in humans [2,3] and 3.7–15% in dogs [4,5,6]. This viper is endemic and a leading venomous snakebite agent in the Mediterranean area, including Western Syria, Northwestern Jordan, Northern and Central Israel, the Palestinian Authority, and Lebanon [7,8,9,10,11,12,13]. Its natural habitat is mostly woodland and scrub, but it is now found in agricultural rural areas, and even in densely populated regions [14].

Its venom is a complex mixture of pharmacologically active molecules, divided into five groups, including viperotoxin, hemorrhagins, angioneurin growth factors, integrin inhibitors, and L-amino acid oxidases [15,16,17,18,19,20,21,22,23,24,25,26,27,28,29,30,31,32,33,34,35,36]. The lethal two-component viperotoxin has neurotoxic and myotoxic activities, resulting from the synergistic action of two proteins, an acidic protein endowed with phospholipase A_2_ (PLA_2_) activity, and a basic protein lacking any known enzymatic activity [19]. Phospholipases hydrolyze fatty acids from glycerophosphatides and are classified into four subtypes, A to D (types C and D are considered phosphodiesterases). Phospholipases A catalyze the hydrolysis of one ester bond in 1,2-diacyl-sn-glycero-3-phosphatides and have positional specificity. Those hydrolyzing the glycerol moiety 1 and 2 position bonds are designated A_1_ and A_2_, respectively. Phospholipase B or lysophospholipase catalyzes the hydrolysis of monoacyl-phosphatides [20]. 

The local and systemic clinical symptoms of *Dp* envenomation result from the synergistic pharmacological activity of the enzymatic and non-enzymatic venom proteins, inducing increased capillary permeability, endothelial damage, platelet aggregation and dysfunction, thromboplastin and thrombin inhibition, factors X and V activation, neutrophilia, leukocytosis, thrombocytopenia, increased fibrinolysis and hypofibrinogenemia, decreased protein C activity, histamine and kinin release, and several presynaptic neurotoxic effects [4,37,38,39,40]. 

The clinical signs of the envenomation in dogs vary depending on the site [4,5], individual patient susceptibility, and the volume of injected venom, which does not necessarily correlate with prey size [15]. In dogs, the clinical manifestations are mostly local, i.e., around the envenomation site, although systemic signs are not uncommon [5]. The most common local signs, depending on the envenomation site, include swelling and edema, viper fang penetration marks, hypersalivation, lameness, bleeding tendency, and local petechiae [5]. Systemic signs include tachypnea, panting, hyperthermia, tachycardia, dyspnea, lymphadenomegaly, and mental status abnormalities, ranging from mental dullness to coma [5,6]. 

The common hematological abnormalities in envenomed dogs include hemoconcentration, leukocytosis, and thrombocytopenia [5,6], while frequent serum chemistry abnormalities include increased muscle enzyme activity (i.e., creatine kinase (CK), lactate dehydrogenase (LDH), and aspartate transaminase (AST)), hypertriglyceridemia, mild hyperglycemia, hyperbilirubinemia, hyperglobulinemia, and hypocholesterolemia [5,6]. The risk factors for death of this envenomation in dogs include limb envenomation, nocturnal bites, dog body weight < 15 kg, occurrence of severe lethargy, hypothermia, systemic bleeding, shock, dyspnea or tachycardia upon admission, thrombocytopenia, hemostatic abnormalities, and development of venom-induced consumptive coagulopathy (VICC) [4,5,6]. 

In humans naturally envenomed by *Dp*, and in rabbits injected with *Dp* venom, transient hypocholesterolemia was noted (dose-dependent in the latter), suggesting that serum cholesterol concentration (sChol) is inversely correlated with the envenomation severity [7]. We therefore hypothesized that sChol, upon admission, might serve as a surrogate marker of the severity and prognosis of *Dp* envenomation in dogs as well, potentially aiding in its assessment and in therapeutic decision-making. The aims of this study were to determine the sChol of dogs envenomed by *Dp* and its associations with clinical and other laboratory findings upon admission to the hospital, with the envenomation severity, and with death. The secondary aim was to look for other markers of the severity and outcome of this envenomation upon admission to the hospital.

## 2. Results

The study included 415 dogs, with a median age of 60 months (range; 3–276) and a median body weight of 25.6 kg (range; 1.8–77.0). Most dogs (359/415; 87%) were envenomed during spring and summer (March to August) (Figure 1). Selected clinical parameters and laboratory analytes upon admission to the hospital are summarized in Table 1. The death rate was 11% (47/415).

The heart rate (HR) upon admission was associated (*p* < 0.0001) with survival (median HR, 140 bpm (range, 50–280) and 180 bpm (range, 48–240) of survivors and non-survivors, respectively) (Table 1; Figure 2). The proportion of tachycardia (HR > 139 bpm) was higher (*p* = 0.008) among the non-survivors compared to the survivors (Table 1).

Signs of bleeding or hematoma upon admission were more common (*p* = 0.003) among the non-survivors (21/41 dogs; 49%) than among the survivors (74/284 dogs; 26%). The mentation status upon admission differed significantly between the outcome groups, with normal mentation noted in 61% and 16% of the survivors and non-survivors, respectively. Among the non-survivors, severe depression and coma were noted in 44% and 22%, respectively, compared to 9% and 3% among the survivors, respectively (*p* < 0.0001). 

Circulatory shock upon admission was more common (*p* < 0.0001) among the non-survivors (65%) compared to the survivors (27%). There were no significant group differences in occurrence of swelling (mild to severe) and lag of time from envenomation to admission (i.e., <12 h or >12 h post-envenomation). 

Upon admission, sChol (reference interval [RI], 135–361 mg/dL) was lower (*p* = 0.026) among the non-survivors (median, 128.0 mg/dL; range, 10.7–376.0) than among the survivors (166.0 mg/dL; range, 23.0–450.7) (Table 1; Figure 3a); hypocholesterolemia (sChol < 135 mg/dL) was more prevalent (*p* = 0.026) amongst the non-survivors (Table 1). The platelet count (RI, 143–400 × 10^3^/µL) was lower (*p* < 0.0001) among the non-survivors (median, 99 × 10^3^/µL; range, 13–341 × 10^3^/µL) than among the survivors (median, 187 × 10^3^/µL; range, 0–1826 × 10^3^/µL) (Table 1; Figure 3b); and thrombocytopenia (platelet count <143 × 10^3^/µL) was more common (*p* < 0.0001) amongst non-survivors (Table 1). Serum albumin concentration (sAlb; RI, 3.0–4.4 g/dL) was lower (*p* = 0.032) among the non-survivors (median, 3.19 g/dL; range, 1.3–4.9) than among the survivors (median, 3.5 g/dL; range, 1.1–5.3) (Table 1; Figure 3c); and hypoalbuminemia (sAlb < 3 g/dL) occurred more frequently among the non-survivors (*p* = 0.003) (Table 1). Serum creatinine concentration (sCr) upon admission among the non-survivors (median, 1.3 mg/dL; range, 0.5–4.8) was higher (*p* < 0.0001) than among the survivors (median, 1.1 mg/dL; range, 0.3–4.2) (Table 1; Figure 3d); and azotemia (sCr > 1.2 mg/dL) was more prevalent amongst the non-survivors (*p* < 0.0001) (Table 1). 

sChol and sAlb, upon admission, were moderately, significantly, and positively correlated (*r_s_* = 0.557; *p* < 0.0001). sChol was weakly, significantly, and negatively correlated with the length of hospitalization (*r_s_* = −0.258; *p* < 0.0001) and HR (*r_s_* = −0.146; *p* = 0.004), and weakly, significantly, and positively with the platelet count (*r_s_* = 0.245; *p* < 0.0001) and total plasma protein concentration (TPP) (*r_s_* = 0.440; *p* < 0.0001). sChol was lower (*p* < 0.0001) in dogs admitted to the hospital >12 h post-envenomation (median, 110 mg/dL; range, 11–214) than in those presented earlier (median, 172 mg/dL; range, 23–451). Among those admitted >12 h post-envenomation, hypocholesterolemia occurred in 81% of the dogs, and all non-survivors admitted >12 h post-envenomation showed hypocholesterolemia (median sChol, 59 mg/dL; range, 11–124). With every 1 mg/dL decrease in sChol, the risk ratio of death increased by 1.004 (95% confidence interval, 1.000–1.009).

## 3. Discussion

*Daboia (Vipera) palaestinae* envenomation in dogs, a medical emergency associated with considerable morbidity and mortality, occurs commonly in Israel [4,5,6]. In this large cohort of dogs, sChol concentration upon admission was lower among the non-survivors than among the survivors, and lower in those presented >12 h post-envenomation than in those presented earlier. In humans, only two studies investigated sChol in snakebite victims; the severity of *Dp* envenomation was associated (*p* < 0.0001) with sChol in 44 human patients when comparing patients with mild, moderate, and severe clinical envenomation manifestations (sChol mean {SD}, 175 {49}, 137 {36}, and 96 {40} mg/dL, respectively) [7]. Another study examined sChol within 24 h of admission and with 10-h overnight fasting in 205 consecutive elapid or viperid snakebite victims. Among the 146 victims classified with serious envenomation, 116 (79%) had sChol ≤ 150 mg/dL. Overall, among the hypocholesterolemic patients, 116 (78%) sustained serious envenomation. The relative risk of moderate to severe envenomation was 2.7-fold in hypocholesterolemic patients compared to normo- and hypercholesterolemic patients [41]. 

Crude viper venom contains proteolytic enzymes, phospholipase A_2_, hyaluronidase, and phosphoesterases [42], inducing endothelial lesions and capillary leakage [43], possibly leading to extravasation of plasma and plasma protein and other molecules, thereby decreasing sChol due to lipoprotein leakage [7]. This pathogenesis might be supported by the concomitant sAlb decrease and the significant, positive correlation between sAlb and sChol noted herein. Nevertheless, when rabbits were injected with a low *Dp* venom dose [7], a significant decrease in sChol was noted, far in excess of the concomitant change in sAlb. The authors, therefore, hypothesized that mechanisms other than change in sAlb possibly play a role in this decrease in sChol during severe *Dp* envenomation [7]. 

Low sChol following severe *Dp* envenomation possibly reflects increased breakdown of cholesterol (and other lipid fractions), attributable to the direct effect of snake venom enzymes, most likely of the phospholipase family [41]. This hypothesis is in line with previous findings, where rabbits injected with isolated *Dp* venom fractions showed the greatest sChol decrease when the PLA_2_-containing one was injected (21%) compared to injections of the hemorrhagic fraction, containing the proteolytic enzymes (7.5%) and the non-toxic fraction (0%) [7]. 

Additionally, low-density lipoprotein (LDL) uptake by macrophages is enhanced following oxidative modification by PLA_2_, converting LDL to a form recognized by macrophages, and greatly enhancing cellular cholesterol-ester uptake and accumulation [44,45]. Similarly, human endothelial cell and hepatoma cell (HepG2) exposure to PLA_2_-modified LDL or high-density lipoprotein (HDL) enhanced lipid deposition [46,47]. PLA_2_-treated LDL binding to human adipocytes is enhanced, suggesting increased affinity of lipoproteins containing hydrolyzed phospholipid to the HDL/LDL receptor, which mediates both HDL and LDL binding and cholesterol delivery to adipocytes [48]. These experimental data suggest that all PLA_2_-modified lipoprotein classes possibly exhibit increased capacity to transfer cholesterol to vascular macrophages and other nonvascular tissue cells. 

Patients with systemic infectious diseases (e.g., sepsis) or burns might show decreased sChol along with decreased HDL_C_ and LDL_C_ concentrations. Moreover, it is well-documented that serum PLA_2_ activity is significantly elevated during such diseases, characteristic of a positive acute-phase protein [49]. Patients with certain chronic diseases (e.g., metastatic tumors and infections) might display hypocholesterolemia. In these conditions, serum proinflammatory cytokine (e.g., interleukin [IL]-1β, IL-6, tumor necrosis factor [TNF]-α, and interferon-γ) concentrations increase, and these cytokines possibly act as PLA_2_ inducers, which increases the PLA_2_ secretion rate by hepatocytes and other cells. Increased serum PLA_2_ activity has been suggested to lead to increased reticulohistiocytic system lipoprotein clearance, mainly through the liver, ultimately resulting in hypocholesterolemia [49]. This hypothesis is supported by a study where hypercholesterolemic rabbits, treated by an extracorporeal circuit containing immobilized snake venom PLA_2_, showed a significant decrease in sChol. Notably, PLA_2_-treated LDL was removed from the rabbits’ bloodstream up to 17-fold faster than their native LDL, and the liver was identified as the primary organ responsible for this enhanced LDL plasma clearance [50]. 

An alternative hypothesis suggests that cholesterol might be directed to increased steroidogenesis, as a secondary reaction to snakebite envenomation, leading to decrease in sChol [41,51]. 

Interestingly, herein, sChol was lower in dogs admitted to the hospital >12 h post-envenomation than in those admitted earlier. This was possibly a combination of more severe and prolonged plasma extravasation due to more progressive tissue damage, alongside longer snake venom PLA_2_ impact on cellular cholesterol uptake. 

Hypoalbuminemia was also significantly more common amongst the non-survivors than among survivors. Hypoalbuminemia occurs in dogs, horses, and humans envenomed by *Dp*, possibly due to albumin leakage and extravasation secondary to vasculitis and capillary damage, especially at the envenomation site [5,7,52]. Additional mechanisms accounting for this hypoalbuminemia might include systemic inflammation, as albumin is a negative acute-phase protein [53], and albumin renal loss, as proteinuria occurs in both dogs and humans envenomed by viperid snakes [54,55]. 

As previously described in dogs and humans [5,56], in this large cohort, thrombocytopenia was significantly more pronounced and common among the non-survivors than among the survivors. Thrombocytopenia possibly occurs due to extravasation, severe vasculitis with bleeding, platelet sequestration in inflamed tissues, and formation of VICC [5]. The prothrombin time (PT) and activated partial thromboplastin time (aPTT) upon admission were similar among the survivors and non-survivors in this study. Consequently, their clinical implication and the possible occurrence of VICC upon admission are hard to assess, as in this study, while repeated clotting times measurement during hospitalization, fibrinogen, fibrin/ogen degradation product and D-dimer concentrations, and antithrombin and protein C activity were not measured in most dogs, upon admission, and later on, precluding assessment for presence of VICC. In previous studies of *Dp* envenomation in dogs, 24 h post-presentation, both the PT and aPTT were prolonged and were significantly associated with death [5,6].

In this study, sCr upon admission was higher among the non-survivors than among the survivors. Acute kidney injury (AKI) is associated with snakebites in both dogs and humans [57,58], primarily caused by the nephrotoxic effects of myoglobinuria and hemoglobinuria secondary to rhabdomyolysis, VICC, toxic nephropathy, hemodynamic instability (i.e., hypovolemic and distributive shock) with renal ischemia, and ischemic necrosis, mediated by both vasoconstriction and procoagulant microthrombotic effects [57,58]. Numerous studies in humans and dogs show that the development of AKI during hospitalization is associated with increased morbidity and mortality rates, and dogs envenomed by *Dp* are no different in that matter [59,60]. 

The associations of tachycardia, higher HR, bleeding, signs of circulatory shock, and abnormal mentation status upon admission with death noted in this study are likely associated with hemodynamic instability, a known risk factor of death in *Dp* and other viper envenomations [5,7,61]. Shock in this envenomation possibly occurs due to acute hypersensitivity reaction (i.e., anaphylaxis) or direct venom actions (e.g., vasculitis, blood component extravasation, microthrombi, and inflammatory mediators) [5,7,61]. 

This retrospective study has several inherent limitations. First, the cohort size, although the largest yet, and especially the number of non-survivors, is nevertheless limited, and this, with some missing data in the medical records, weakened the statistical analyses. Second, this study included dogs admitted to a single teaching hospital over a very long period, during which improvements in diagnostic and therapeutic modalities have occurred. This very likely introduced variance and possibly influenced the clinical outcome of certain individuals. Our results should therefore be applied cautiously to other clinical settings. During this long study period, several chemistry analyzers with several different reagents for total sChol measurement were used, which possibly introduced variance. Nevertheless, all reagents for total cholesterol measurement used herein are based on the Abell–Levy–Brodie–Kendall (i.e., Abell–Kendall; AK) reference measurement procedure. It has been shown that this method is precise and stable, and the results of different analyzer and reagent manufacturers are highly correlated over a long observation period [62]. Third, five non-survivor dogs showing deterioration, despite ongoing efforts, were euthanized at their owners’ request, and financial constraints might have influenced this decision, which possibly affected the outcome group comparisons. Financial constraints possibly also limited the performance of some laboratory tests upon admission, which contributed to missing data. Financial constraints always somewhat limit small animal diagnosis and treatment, and this limitation is unavoidable in the routine clinical setting and is a common limitation in retrospective studies of dogs. 

## 4. Conclusions

In dogs envenomed by *Dp*, sChol upon admission to the hospital is a marker of the severity and outcome of the envenomation and is lower in dogs presented late (>12 h) post-envenomation. There is a moderate positive correlation between sChol and sAlb upon admission. The HR, platelet count, sAlb, and sCr, upon admission, are also potential markers of the severity and outcome of *Dp* envenomation. Future studies examining the trends of change of these markers during hospitalization and their associations with morbidity and mortality are warranted.

## 5. Materials and Methods

### 5.1. Selection of Dogs and Data Collection

The medical records of dogs admitted to the Veterinary Teaching Hospital (HUVTH) between 1989 and 2020 and diagnosed with *Dp* envenomation were retrospectively reviewed. Envenomation was diagnosed based on ≥2 of the following: (1) envenomation occurred in geographic areas where *Dp* is the sole venomous snake; (2) the biting snake was identified as *Dp* by dog owners or HUVTH clinicians; (3) characteristic *Dp* fang penetration marks were identified by HUVTH clinicians; (4) the clinical signs were of acute onset, and were typical of *Dp* envenomation in dogs [5,6], occurring in animals that had been clinically normal before the envenomation had occurred. sChol was measured upon admission to the HUVTH in all dogs in this cohort. Dogs treated with *Dp* antivenom or blood products prior to the admission to the HUVTH were excluded. 

Data collected from the medical records included the signalment, date of admission, the lag of time from envenomation to admission, the medical history, clinical findings (e.g., vital signs, mental status, bleeding evidence, and envenomation site), laboratory findings upon admission, the number of antivenom units administered per dog, and the length of hospitalization and the final outcome (i.e., survival to discharge from the hospital, or in-hospital death, or euthanasia due to clinical deterioration, despite ongoing therapy). Upon admission to the hospital, emergency clinicians subjectively classified the hemodynamic status of dogs as in shock or in the absence of shock.

### 5.2. Collection of Blood and Laboratory Methods 

Over the very long observation period of this study, several hematology and chemistry analyzers were used. Blood samples for complete blood count (CBC) were collected in potassium-EDTA tubes and analyzed (Minos ST-Vet, Minos, Montpelier, France; Arcus, Abacus or Abacus Junior Vet, Diatron, Vienna, Austria; Advia 120 or 2120i, Siemens, Erfurt, Germany) within 30 min from collection. The packed cell volume (PCV) was measured manually by centrifuging whole blood in heparinized capillaries. Total plasma protein concentration was measured via refractometry (clinical refractometer, Atago, Tokyo, Japan). Blood samples for serum chemistry were collected in tubes with no anticoagulant; and with gel separators, allowed to clot, were centrifuged within 60 min from collection; and harvested sera were analyzed (Kone Progress Selective Chemistry Analyzer, Kone Corporation Instrument Group; Espoo, Finland; Maxmat SA PL, Maxmat, Montpellier, France; Cobas-Mira, Cobas Integra 400 Plus and Cobas 6000, Roche, Mannheim, Germany; at 37 °C) immediately, or stored at 4 °C pending analysis, performed within 12 h from collection. Blood samples for PT and aPTT were collected in 3.2% trisodium citrate tubes and centrifuged within 15 min from collection. Harvested plasma was then analyzed immediately (KC 1A micro, Amelung, Lemgo, Germany; ACL-200, ACL-9000 and ACL Top 300, IL, Milano, Italy).

### 5.3. Statistical Methods 

The distribution pattern of continuous variables was determined using the Shapiro–Wilk’s test. These variables were compared between 2 groups using the Student’s *t*-test or the Mann–Whitney’s U-test, depending on the data distribution pattern. Categorical variables were compared between groups using the chi-square or Fisher’s exact tests as appropriate. Relations between two continuous variables were determined via Pearson or Spearman correlations, depending on the data distribution pattern. All tests were 2-tailed. In all, *p* < 0.05 was considered significant. Analyses were performed using a statistical software package (SPSS 28.0, IBM, Armonk, NY, USA).

## Figures and Tables

**Figure 1 toxins-15-00609-f001:**
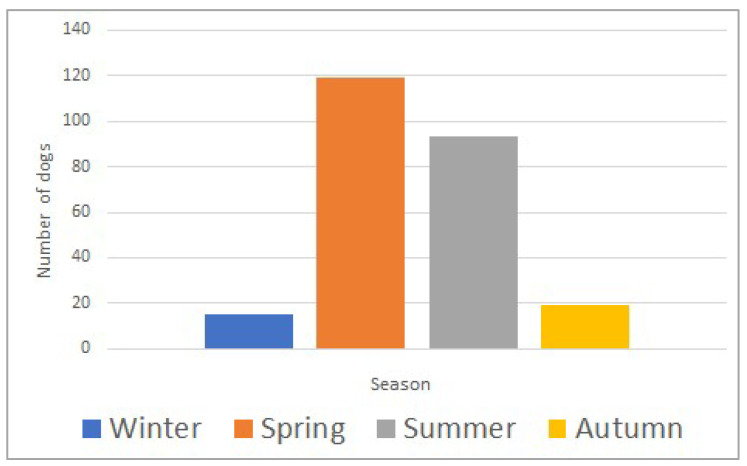
Seasonal distribution of *Daboia plaestinae* envenomation in 415 dogs (winter: December–February; spring: March–May; summer: June–August; autumn: September–November).

**Figure 2 toxins-15-00609-f002:**
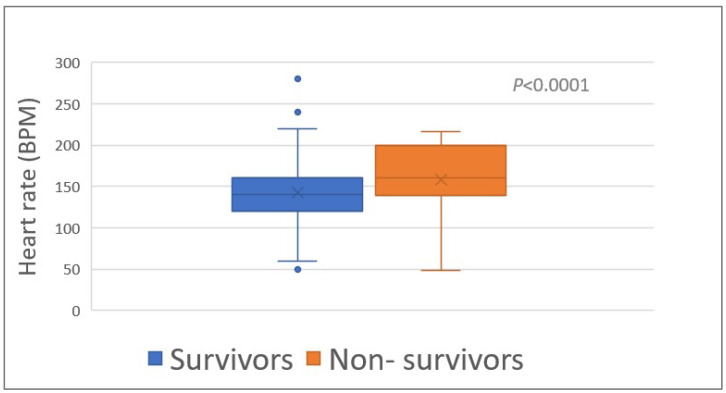
Heart rate of 348 survivors and 46 non-survivors of *Daboia palaestinae* envenomation upon admission to the hospital. BPM, beats per minute.

**Figure 3 toxins-15-00609-f003:**
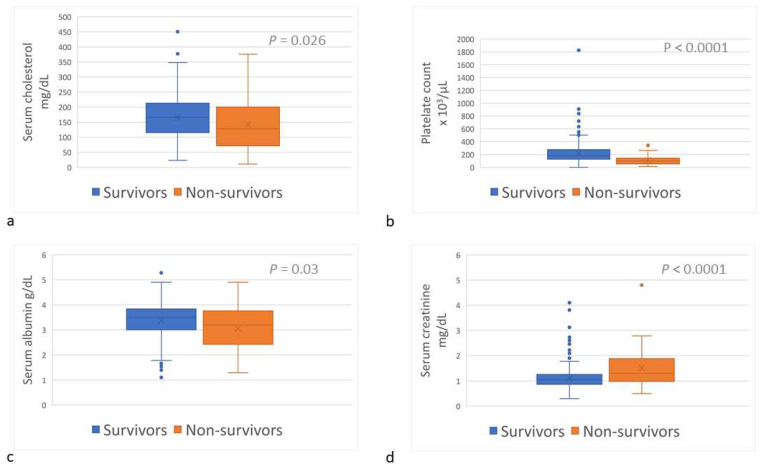
Serum cholesterol concentration (**a**), platelets count (**b**), serum albumin concentration (**c**), and serum creatinine concentration (**d**) of survivor and non-survivor dogs of *Daboia palaestinae* envenomation upon admission to the hospital.

**Table 1 toxins-15-00609-t001:** Selected clinical parameters and laboratory analytes upon admission in 415 dogs presented to a veterinary emergency and critical care unit for acute *Daboia palestinae* envenomation.

	All Dogs					Survivors				Non-Survivors				*p* ^1^	*p* ^2^
Analyte	RI	N (%)	Median (Range)	N (%) <RI	N (%) >RI	N	Median (Range)	N (%) <RI	N (%) >RI	N	Median (Range)	N (%) <RI	N (%) >RI		
Rectal temperature (°C)	38–39.2	373 (90)	38.8 (35.4–41.0)	36 (10)	110 (30)	332	38.8 (35.4–41.0)	30 (9)	95 (29)	41	38.7 (37.3–40.1)	6 (15)	15 (37)	0.559	0.255
Respiratory rate (breaths/min)	10–40	369 (89)	100 (16–160)	1 (0.3)	143 (39)	330	100 (16–146)	0 (0)	117 (36)	39	60 (24–160)	1 (3)	26 (67)	0.853	0.014
Heart rate (bpm)	60–139	394 (95)	140 (48–280)	2 (0.5)	266 (66)	348	140 (50–280)	1 (0.3)	227 (65)	46	180 (48–240)	1 (2)	39 (85)	<0.0001	0.008
Body weight (Kg)	NA	376 (91)	25.5 (1.8–77.0)	NA	NA	332	26.0 (1.8–71.0)	NA	NA	44	24.0 (3.0–77.0)	NA	NA	0.468	NA
Leukocytes (×10^3^/µL)	5.2–13.9	397 (96)	15.1 (2.8–70.75)	6 (2)	228 (57)	353	15.1 (4.1–70.75)	3 (1)	207 (59)	44	14.0 (2.8–40.2)	3 (7)	21 (48)	0.855	0.007
Packed cell volume (%)	37–56	401 (97)	53 (15–77)	21 (5)	135 (34)	356	53 (16–77)	19 (5)	115 (32)	45	55 (15–77)	2 (4)	20 (44)	0.203	0.292
Red blood cells (×10^6^/µL)	5.7–8.8	386 (93)	7.8 (1.16–11.8)	20 (5)	64 (17)	344	7.8 (2.7–10.8)	16 (5)	52 (15)	42	8.2 (1.16–11.8)	4 (10)	12 (29)	0.505	0.03
Total plasma protein (gr/dL)	5.4–7.5	407 (98)	6.4 (2.4–10.5)	65 (16)	44 (11)	361	6.4 (2.4–10.5)	52 (14)	39 (11)	46	6.0 (3.0–9.0)	13 (28)	5 (11)	0.152	0.057
Platelets (×10^3^/µL)	143–400	392 (94)	174 (0–1826)	131 (33)	35 (9)	348	187 (0–1826)	98 (28)	35 (10)	44	99 (13–341)	33 (75)	0 (0)	<0.0001	<0.0001
Serum cholesterol (mg/dL)	135–361	415 (100)	163.6 (10.7–450.7)	153 (37)	3 (0.7)	368	166 (23–450.7)	128 (35)	2 (0.5)	47	128 (10.7–376)	25 (53)	1 (4.2)	0.026	0.026
Serum albumin (gr/dL)	3.0–4.4	390 (94)	3.4 (1.1–5.3)	108 (28)	17 (4)	346	3.50 (1.1–5.3)	87 (25)	14 (4)	44	3.19 (1.3–4.9)	21 (48)	3 (7)	0.032	0.003
Serum creatinine (mg/dL)	0.3–1.2	407 (98)	1.07 (0.3–4.8)	NA	137 (34)	362	1.05 (0.3–4.2)	NA	110 (30)	45	1.3 (0.5–4.8)	NA	27 (60)	<0.0001	<0.0001
Activated partial thromboplastin time (sec)	11–17.4	347 (84)	15.7 (7.35–100)	74 (21)	192 (55)	304	15.7 (7.35–100)	69 (23)	171 (43)	43	15.7 (8.6–58)	5 (12)	21 (49)	0.219	0.242
Prothrombin time (sec)	6–8.4	346 (83)	8.6 (5.0–100)	8 (2)	181 (53)	304	8.5 (5.0–100)	7 (2)	155 (51)	42	9.4 (5.7–26.3)	1 (2)	26 (62)	0.099	0.295

RI, reference interval; N, number of available cases or analyses; (%), percent of cases/analyses of all cases/analyses in the group; NA, not applicable. ^1^ Groups compared using the Student’s *t*-test or Mann–Whitney U-test. ^2^ Groups compared using the chi-square or Fisher’s exact tests. *p* < 0.05 is considered significant.

## Data Availability

The row dataset includes some confidential information, including medical record number and data of admission; hence, it will not be freely available. Nevertheless, other data can be obtained by contacting the corresponding author. The data presented in this study are available on request from the corresponding author. The data are not publicly available due to privacy restrictions.

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
