# Peer review of "Serum Cholesterol Concentration on Admission in 415 Dogs Envenomated by Daboia (Vipera) palaestinae as a Marker of Envenomation Severity and Outcome—A Retrospective Study"

_toxins, 2023, doi:10.3390/toxins15100609_

Round 1

Reviewer 1 Report

Thank you for sending me a very interesting manuscript to review. I was intrigued by the topic of your study and very pleased with the whole article. It is clear from the works cited (although the article is sent without the names of the authors) that you have been working on this topic for a long time and from different angles. Venomous snakes have always received a lot of attention in research, and I too am personally interested in the composition and effects of venoms, even if it is not the main focus of my research. Most studies focus, for obvious reasons, on humans. After all, snakebite is one of the real threats to human health and life even today. That's why I was pleased that your work focused on other animals, specifically dogs. For example, with the more toxically important tarantulas, the reaction to their venom in humans and dogs can be very different. In humans, it can manifest itself only in local and short-term symptoms, whereas in dogs it is more likely to cause death. So I was wondering what the situation is with snake venom. In addition, the risk of death of pets (mostly dogs) related to snakebites is one of the factors that influence people's negative attitude towards snakes as a group. Therefore, better knowledge regarding the incidence and effects of venom can also help in education campaigns for conservation projects.

As far as the text of the article is concerned, I have no serious objections. Of course, one could think of comments on what information to add and from what point of view to comment on your results, or how to better present your results. But in my opinion, everything is sufficiently described and the article reads well. I wish you every success in your future work.

Author Response

We would kindly like to thank the reviewer for their kind words. We appreciate the time they took to review our manuscript.

Reviewer 2 Report

The study analyzes the laboratory parameters of 415 dogs bitten by Daboia palaestine, admitted to a Veterinary Teaching Hospital between 1989 and 2020.

The authors observed a significant decrease in serum cholesterol in animals that died compared to survivors. They proposed that this parameter could be a marker of the severity of the envenoming.

However, the authors also show a significant drop in serum albumin levels and thrombocytopenia in cases that resulted in the death of the animals.

It seems more appropriate to point out this general picture (drop in serum cholesterol, albumin, and thrombocytopenia) as markers of severity rather than attributing this role to serum cholesterol alone.

Overall, the work is well written, presents important data for diagnosing the severity of snake bites, and fits within the scope of Toxins.

Some specific points:

Figures 2 and 3 are unnecessary as they replicate data already presented in Table 1.

In line 158, the content in parentheses is unnecessary, but if you keep it, change the bracket symbols for standard deviations, as they are confused with bibliographic citations.

The bibliographic citations could be up-to-dated.

Author Response

Comment: The study analyzes the laboratory parameters of 415 dogs bitten by Daboia palaestine, admitted to a Veterinary Teaching Hospital between 1989 and 2020. The authors observed a significant decrease in serum cholesterol in animals that died compared to survivors. They proposed that this parameter could be a marker of the severity of the envenoming. However, the authors also show a significant drop in serum albumin levels and thrombocytopenia in cases that resulted in the death of the animals. It seems more appropriate to point out this general picture (drop in serum cholesterol, albumin, and thrombocytopenia) as markers of severity rather than attributing this role to serum cholesterol alone.
Response: This study's main aim was to explore serum cholesterol concentration upon admission as a marker of the severity of envenomation in dogs, as this has never been investigated. The secondary aim was to look for additional possible markers of the severity of envenomation, some of which have already been described previously, but not in such a large-scale cohort of dogs. Having identified serum albumin and creatinine concentrations and thrombocytopenia upon admission as such markers, we believe we have consistently emphasized this point throughout the manuscript:

  1. In the Results section, we initially describe the significance of cholesterol (main aim of the study), albumin, platelet count, and creatinine concentrations (secondary aims of this study) in relation to survival (Lines 118-133), and emphasize this point with Figure 3.
  2. We further underscore the correlation between cholesterol and albumin, platelet count, and creatinine concentration in the Results section (Lines 139-143).
  3. In the Discussion section, we address the relationship between serum cholesterol and serum albumin concentrations (Lines 213-219), and provide a whole separate paragraph to the importance of thrombocytopenia and azotemia in the prognosis of envenomation (Lines 222-240).
  4. Finally, in our Conclusions, we highlight the potential significance of these parameters as markers of envenomation severity, which was demonstrated in previous studies. The novelty of our present study lies in emphasizing the importance of serum cholesterol concentration upon admission as a potential marker of severity and outcome of this envenomation, and we wanted to present it as a standalone focus.

Comment: Overall, the work is well written, presents important data for diagnosing the severity of snake bites, and fits within the scope of Toxins.
Response: Thank you.

Some specific points:

Comment: Figures 2 and 3 are unnecessary as they replicate data already presented in Table 1.
Response: We believe that Figures 2 and 3 emphasize the significance of the presented parameters and provide the reader with a clearer understanding of the most prominent and important results of the study. This issue can be left to the Editor's discretion. Nevertheless, should the reviewer insist, or should the Editor request, these figures can be omitted.

Comment: In line 158, the content in parentheses is unnecessary, but if you keep it, change the bracket symbols for standard deviations, as they are confused with bibliographic citations.
Response: Done.

Comment: The bibliographic citations could be up-to-date.
Response: We are aware that many studies of the composition and action of D. palaestinae venom have been made between the 1950s and 1970s. Unfortunately, there are no current or more recent studies on this subject, to the best of our knowledge. Nevertheless, we would gladly accept specific suggestions of suitable, relevant more up-to-date (or preferably, recent) articles that might potentially provide further support for our manuscript.

Reviewer 3 Report

In this paper, the authors use 30 years of historical data from veterinary hospital dogs to show that serum cholesterol levels are a marker of the severity of Daboia palaestinae bites. Although the data is very interesting, the paper would be easier to understand for readers if the results and discussion were improved, so I would like to see the following points corrected.

1. Regarding Table 1: Please write in the margin what "N (%)" in Table 1 means. I don't know what the numbers written in the table represent.

2. Regarding Table 1: It seems that there are some missing values in the data for each item listed in Table 1, but please write the number of missing values for each item in the margin.

3. Regarding discussion: An equine antitoxin for the treatment of Daboia palaestinae is commercially available in Israel. Wouldn't this antivenom be administered as a treatment to the dogs that were bitten by the poisonous snakes reported by the authors? If you include data on dogs treated with antitoxin, please state this in your discussion and consider the relationship between the therapeutic effect and serum cholesterol concentration.

Author Response

Comment 1. Regarding Table 1: Please write in the margin what "N (%)" in Table 1 means. I don't know what the numbers written in the table represent.
Response: Done.

Comment 2. Regarding Table 1: It seems that there are some missing values in the data for each item listed in Table 1, but please write the number of missing values for each item in the margin.
Response: Thank you for this comment. We have amended Table 1, and have added the percent of available cases/analyes for each parameter of all We believe that stating the total number of patients for whom the relevant analyte was measured, along with the corresponding numbers for both survivors and non-survivors, along with the percentages (which represent the actual number of available analyses of the total number of dogs in this study (i.e., 415 dogs), and this will actually provide the reader the data the reviewer seeks). The footnotes have been amended as well.

Comment 3. Regarding discussion: An equine antitoxin for the treatment of Daboia palaestinae is commercially available in Israel. Wouldn't this antivenom be administered as a treatment to the dogs that were bitten by the poisonous snakes reported by the authors? If you include data on dogs treated with antitoxin, please state this in your discussion and consider the relationship between the therapeutic effect and serum cholesterol concentration.
Response: An equine-based Daboia palaestinae antitoxin has been and still is available in Israel. and is sometimes used in treatment of enveomated dogs. In fact, at present, in the last 10 years, there are two different available antivenoms, and since last year, a third antivenom has become available as well. Nevertheless, we have specific reasons for excluding a discussion of the use of antivenom in our patients. This study's main aim was to describe serum cholesterol concentration on admission as a severity and outcome marker of the envenomation. It should be stressed that serum cholesterol upon admission to the hospital did not play a part in the decision whether to administer antivenom in this cohort, since the attending clinicians did not have in real time the database and the statistical analyses which were made in this study. While it would be interesting to investigate serum cholesterol concentration after treatment, and throughout hospitalization, in the field of veterinary medicine, financial constrains very often limit costly therapy (e.g., antivenom) and repeated laboratory analyses (e.g., repeated serum chemistry analyses). One unit (10 mL) of D. palaestinae antivenom costs 1500 USD, which is most often cost-prohibitive for dog owners (unlike in human patients). The decision whether to administer it is therefore biased, depending on the owner's financial means, and is not necessarily made solely on the clinical severity or laboratory derangements. This financial aspect does introduce serious confounding factors.